# When Critical View of Safety Fails: A Practical Perspective on Difficult Laparoscopic Cholecystectomy

**DOI:** 10.3390/medicina59081491

**Published:** 2023-08-19

**Authors:** Catalin Alius, Dragos Serban, Dan Georgian Bratu, Laura Carina Tribus, Geta Vancea, Paul Lorin Stoica, Ion Motofei, Corneliu Tudor, Crenguta Serboiu, Daniel Ovidiu Costea, Bogdan Serban, Ana Maria Dascalu, Ciprian Tanasescu, Bogdan Geavlete, Bogdan Mihai Cristea

**Affiliations:** 1Faculty of Medicine, Carol Davila University of Medicine and Pharmacy Bucharest, 020021 Bucharest, Romania; catalin.alius@umfcd.ro (C.A.); geta.vancea@umfcd.ro (G.V.); ion.motofei@umfcd.ro (I.M.); corneliu.tudor@umfcd.ro (C.T.); crenguta.serboiu@umfcd.ro (C.S.); bogdan.serban@umfcd.ro (B.S.); ana.dascalu@umfcd.ro (A.M.D.); bogdan.geavlete@umfcd.ro (B.G.); bogdan.cristea@umfcd.ro (B.M.C.); 2Fourth General Surgery Department, Emergency University Hospital Bucharest, 050098 Bucharest, Romania; 3Faculty of Medicine, University “Lucian Blaga”, 550169 Sibiu, Romania; dan.bratu@ulbsibiu.ro (D.G.B.);; 4Department of Surgery, Emergency County Hospital Sibiu, 550245 Sibiu, Romania; 5Faculty of Dental Medicine, Carol Davila University of Medicine and Pharmacy Bucharest, 020021Bucharest, Romania; laura.tribus@umfcd.ro; 6Department of Internal Medicine, Ilfov Emergency Clinic Hospital Bucharest, 022104 Bucharest, Romania; 7Third Clinical Infectious Disease Department, Clinical Hospital of Infectious and Tropical Diseases “Dr. Victor Babes”, 030303 Bucharest, Romania; 8Department of General Surgery, Emergency Clinic Hospital “Sf. Pantelimon” Bucharest, 021659 Bucharest, Romania; 9Faculty of Medicine, Ovidius University Constanta, 900470 Constanta, Romania; daniel.costea@365.univ-ovidius.ro; 10General Surgery Department, Emergency County Hospital Constanta, 900591 Constanta, Romania

**Keywords:** laparoscopic cholecystectomy, bile duct injury, acute cholecystitis, anatomical landmarks, critical view of safety, bailout surgery

## Abstract

The incidence of common bile duct injuries following laparoscopic cholecystectomy (LC) remains three times higher than that following open surgery despite numerous attempts to decrease intraoperative incidents by employing better training, superior surgical instruments, imaging techniques, or strategic concepts. This paper is a narrative review which discusses from a contextual point of view the need to standardise the surgical approach in difficult laparoscopic cholecystectomies, the main strategic operative concepts and techniques, complementary visualisation aids for the delineation of anatomical landmarks, and the importance of cognitive maps and algorithms in performing safer LC. Extensive research was carried out in the PubMed, Web of Science, and Elsevier databases using the terms ”difficult cholecystectomy”, ”bile duct injuries”, ”safe cholecystectomy”, and ”laparoscopy in acute cholecystitis”. The key content and findings of this research suggest there is high intersocietal variation in approaching and performing LC, in the use of visualisation aids, and in the application of safety concepts. Limited papers offer guidelines based on robust data and a timid recognition of the human factors and ergonomic concepts in improving the outcomes associated with difficult cholecystectomies. This paper highlights the most relevant recommendations for dealing with difficult laparoscopic cholecystectomies.

## 1. Introduction

Although the pioneers of laparoscopic surgery were initially vociferously opposed by their fellow surgeons, the unequivocal advantages derived from their work have changed many old paradigms [1,2,3]. Laparoscopic cholecystectomy (LC) is one of Europe’s most commonly performed surgical operations, highlighting the significance of the prevalence of biliary iatrogenic complications [4,5]. Performing an LC in acute cholecystitis (AC) was a relative contraindication 20 years ago mainly due to the high rates of common bile duct (CBD) and vasculo-biliary (VB) iatrogenic lesions when laparoscopy was still an emerging technique [6]. The reported incidence of CBD injuries following LC is 0.3 to 0.7%, three times higher than that following open surgery. This trend continues despite numerous attempts to decrease intraoperative incidents by employing better training, superior surgical instruments, imaging techniques, or strategic concepts, such as Strasberg’s critical view of safety (CVS) [7,8,9,10,11,12].

The misinterpretation of anatomical landmarks is the main cause of intraoperative biliary and vascular lesions [13]. In an analysis of 252 laparoscopic bile duct injuries, Way concluded that “the primary cause of error in 97% of cases was a visual perceptual illusion” mainly caused by erroneous decoding of the local anatomy altered by scarring, inflammation, and variations [14,15]. A study in the US based on an anonymous survey including 3657 surgeons showed that most CBD injuries were caused by experienced surgeons rather than novices, who tended to be more cautious and employed judicious explorations and double-checking during procedures [16]. The authors emphasise the necessity of intraoperative awareness and its reverberations on outcomes. Moreover, iatrogenic bile duct injury (IBDI) secondary to cholecystectomy may significantly affect long-term quality of life and have major morbidities. Furthermore, even after reconstructive surgical treatment, such injuries still reduce long-term quality of life [17,18].

Increased landmark identification accuracy in complicated LC has been achieved using NIR fluorescence cholangiography and augmented reality, anterograde, laterodorsal and bipolar dissection strategies, injections of vital dyes into the gallbladder, etc. In some cases, subtotal cholecystectomy or conversion to open surgery is the safest option. 

We compiledthis narrative review as a natural consequence of intensive research to optimise our practice in a tertiary centre dealing with large volumes of acute LC. Hence, this paper offers a documented practical perspective on strategies to avoid complications in emergency laparoscopic cholecystectomy.

## 2. Standardisation of Techniques for Emergency LC

A large variety of surgical techniques for a single condition is highly indicative of unmet technical needs and unacceptable outcomes. More than 50 techniques for LC have been described over the past 40 years [19]. Cushieri’s cautionary advice related to potential safety concerns caused by the excessive use of LC remains one of the unmet challenges of this operation [20]. Naturally, this was regarded as a consequence of little experience when laparoscopy was in its early days, but the number of iatrogenic lesions remained high despite advancements in surgical instruments and the ubiquitous spread of laparoscopy. This is because the complexity of the attempted laparoscopic cases increased. Acute cases have the highest rates due to local conditions; hence, the standardisation of the surgical technique would seem a logical resolution [21]. Yet, to date, there have been only a few attempts to standardise the technique, all applicable mainly to elective cholecystectomy [22,23]. Wkabayasi et al.’s notable contribution derived from the TG18 guidelines enumerated six “safe steps” in LC for acute cholecystitis [24]. 

Currently, ultrasonography (US) imaging, computer tomography (CT), and magnetic resonance imaging (MRI) play fundamental roles in the diagnosis, management, and pre-operative study of acute cholecystitis. Although the diagnostic criteria for diagnosing acute cholecystitis using US and its diagnostic yield vary in different studies, its low invasiveness, widespread availability, ease of use, and cost-effectiveness make it the first-choice imaging method for the morphological diagnosis of acute cholecystitis. If abdominal US does not provide a definitive diagnosis, CT and MRI are other less commonly used imaging techniques to diagnose acute cholecystitis [25,26,27]. MRI and CT can also be used to diagnose gangrenous cholecystitis, whereas only CT is recommended for diagnosing emphysematous cholecystitis [28]. 

MRI or MR cholangiopancreatography (MRCP) allow the accurate study of the anatomy of the biliary system and the evaluation of any anatomical variants and/or accessory ducts, making them pivotal for pre-operative investigation. Using MRI with MRCP in the emergency setting provides rapid, noninvasive, and confident diagnosis, or the exclusion of acute cholecystitis and coexistent choledocholithiasis. Previous literature reports suggest that, when available, MRI should be recommended to provide a prompt and efficient triage of patients with suspected cholecystitis and inconclusive clinical, laboratory, and sonographic findings [29,30].

Iwhasita [31] offers a novel perspective on assessing the difficulty of LC based on objective intraoperative findings grouped into two main categories: factors related to inflammation of the gallbladder (appearance around the gallbladder, in the Calot’s triangle area, or the gallbladder bed and its surroundings) and intra-abdominal factors unrelated to the inflammation, such as visceral fat, liver cirrhosis with collateral vein formation, an anomalous bile duct, physiological adhesion around the gallbladder, or the gallbladder neck mounting on the common bile duct [31].

Sugrue proposed a simplified G10 score that is easy and quick to calculate, suggesting that his instrument might offer easier bailout decisions in difficult cases [32] (Table 1).

The dynamic nature of the inflammatory changes contributes to the difficulty of an acute case, rendering the latter highly dependable on the timing of surgery. Unsurprisingly, both paradigms regarding early and delayed cholecystectomy stem from the same empirical observations. Interval cholecystectomy was preferred when laparoscopy was not universally accessible and highly experienced centres were scarce. The current consensus is that ELC does not cause more iatrogenic complications compared with DLC and that it is best to operate in the first 72 h after presentation [33,34]. Many authors do not specifically refer to surgical techniques for acute cases, presuming that a similar strategy should be employed for emergency, early, and interval cholecystectomies. This is sensible considering that, most of the time, improvisations in difficult cases consist of modifications of traditional steps and concepts used for elective cases. Predictors of difficulty and good timing should be incorporated into a standardised strategic approach to LC for acute cases.

## 3. Anatomical Landmarks and the “Illusion of Form and Shape”

Traditionally established anatomical landmarks taught in medical schools are surprisingly fluid in real-time situations, perhaps because memorisation is meaningless without pattern recognition and contextual understanding. Far from being a philosophical remark, “the illusion of form and shape” has been harshly experienced by all surgeons who have mistaken a CBD for a cystic duct, an error which Archer suggests might be made by almost half of us [21]. The following preventive strategies against anatomical misinterpretation stem from the acknowledgement of “perception heuristics” and the importance of awareness and rules of thumb for structural recognition.

### 3.1. Rouviere’s Sulcus (RS)

Rouviere’s sulcus (RS) is an oblique 2–5 cm groove running to the right of the hepatic hilum above the caudate lobe and containing the right posterior portal pedicle (Figure 1). It accurately indicates the plane of the CBD [35]. A meta-analysis of 4495 cases (cadaveric and laparoscopic) identified a prevalence of 83% of Rouviere’s sulcus with two types of morphology: open and fused [36]. A plane crossing through the sulcus should be above the CBD, rendering dissection anterior to this area safe [37,38,39].

The authors also proposed a transversal plane passing through the R4U line and a vertical plane perpendicular to it (Figure 2). The transversal and vertical planes delineate four zones. Two of the zones are safe for dissection, while the other two are not. This technique of zonal demarcations is useful during intraoperative orientation, bearing in mind that structures in each of the four zones might be pulled towards unsafe zones via either pathological shortening or intraoperative traction (Figure 3).

### 3.2. Divisions of the Subserosal Layer as a Landmark for Safe Dissection

Honda proposed a return to basic histology to understand the dissection technique for the bilaminar subserosal layer surrounding the gallbladder [41]. The subperitoneal tissue is composed of an outer subserosal and an inner subserosal layer. The outer SS layer is thick and fatty and has minimal vascularisation, while the inner SS layer is thin and fibrous and contains arteries and veins in close contact with a shiny surface just above the muscularis propria (Figure 4). A semi-circular cut into the peritoneum followed by dissection under the outer SS layer will expose the shiny inner SS layer and allow for the safe dissection of the gallbladder (Figure 5).

### 3.3. Hepatocystic Triangle and Calot’s Triangle 

When Jean-Francois Calot described this anatomical landmark in his doctoral thesis, he specifically mentioned a rather isosceles triangle delineated between the cystic duct, the CBD, and the cystic artery [42]. The right hepatic artery sometimes accounts for the first part of the upper border. It is unclear when Calot’s triangle became an eponym and when its upper border was replaced with the inferior margin of the liver. This modification, also known as the “Bode Rocko” triangle, has more practical value in laparoscopic anatomy since the variability of the cystic artery is very high [43]. Additionally, the inferior border of the liver constitutes a constant and obvious landmark that is much more useful for laparoscopic anatomy (Figure 6). This differs from the disposition found in cadaveric structures and atlas depictions, mainly because of the stretching that occurs during surgical manoeuvres. The triangle contains the Lund or Mascagni node, fatty and fibrous tissue, and anatomical variations. In patients with a normal BMI, the cystic node marks the emergence of the cystic artery from the right hepatic artery [44].

### 3.4. Anatomical Variations 

Lahey (1948) stated that ‘‘cholecystectomy is a dangerous operation unless one realises that variations are very common’’ [45]. Half of the patients will display anatomical variants of insertion of the cystic duct (CD), with a prevalence of accessory ducts of up to 10% [46]. Gunduz presented a study on MRCPs, demonstrating that the craniocaudal level of the cystic duct’s insertion into the CBD is variable. This leads to confusion, especially when structures are lifted and pulled to gain exposure. [47]. A typical arterial supply of Calot’s triangle was reported in almost 70% of patients, with the cystic artery emerging from the right hepatic artery and coursing through anteromedial and accessory arteries in 7.4% [48]. In a cadaveric study by Dandekar, almost all cystic arteries passed through the hepatobiliary triangle with a thickness ranging from 1 to 5 mm [49]. Although almost 80% of cystic arteries are singular structures, 15% of right hepatic arteries course through the triangle and can be mistaken for a cystic artery [44] (Figure 7 and Figure 8).

Awareness is a dynamic process that contributes to better decision making and optimises surgical behaviour. These well-known anatomical landmarks are sometimes ignored or misinterpreted by surgeons regardless of their experience. From fixation errors to lack of knowledge, multiple cognitive mechanisms are involved in the erroneous decoding of the local anatomy. A culture of safety during LC must admit that **“illusions of forms and shapes**” are real and can be successfully avoided with awareness models and systematic approaches (Table 2).

## 4. Strategic Operative Concepts and Techniques in Acute LC

Lemuel Pran made a pertinent comment in a letter to an editor highlighting that there is confusion in the literature related to what anterograde and retrograde LC approaches signify for various authors. He suggests the avoidance of these terms until worldwide consensus is reached or alternative more intuitive syntagms such as “fundus first” or “Calot first” are used [50].

Pran’s warning about nomenclature inconsistencies seems to be translated, at least partially, into caveats about reporting and counting because “interestingly the flow of bile is bidirectional”. Even the IRCAD recommendations for safe cholecystectomy acknowledge that sometimes a cranial approach, which is generally referred to as “antegrade”, might be labelled as “retrograde” [51]. From this perspective, we will adhere to Pran’s suggestion and use more descriptive terms.

### 4.1. Fundus-First

The technique of fundus-first was introduced due to the need to perform safer LC in patients in whom the classical approach might have precipitated VBI due to inflammatory changes in Calot’s triangle [52]. Jenkins suggests this technique is underused, and its employment could lower the rates of conversions [53]. Conversely, several authors have cautioned against major VBI since the most caudal portion of the cystic plate might be fused with elements of the porta hepatis, secondary to phlogistic conditions [54].

The technique consists of the retraction of the anterior border of the liver followed by dissection from the fundus towards the cystic duct, between the inner and outer layers of the subserosal layer covering the GB. Cranial traction and rotation of the GB allow for the dissection of the cystic duct and artery. This approach is associated with higher blood loss due to the patency of the cystic artery, which is clipped only at the end of the mobilisation. Before any attempt to expose anatomical landmarks, adhesions should be lysed bluntly, sharply, or using electrocautery [55].

### 4.2. Lateral Dorsal Infundibular Approach 

The middle-first approach is a variation of fundus-first used when the GB dome firmly adheres to the anterior border of the liver or when traction is not possible. This technique is based on an attempt to fenestrate between the cystic plate and the GB wall, starting at the angle between the serosa of the GB and the visceral aspect of the liver. Provided that fenestration is achieved, the newly created dissection plane is followed by cephalad with the mobilisation of the GB and then caudad towards the cystic duct. The lateral dorsal infundibular approach is derived from this technique, but dissection begins at a lower level. All of these technique modifications should be employed when the critical view of safety cannot be achieved based on a judicious assessment of the local conditions and available anatomical landmarks. If total cholecystectomy (TC) is not feasible, subtotal cholecystectomy should be the next logical step [56].

### 4.3. Subtotal Cholecystectomy 

The concept of subtotal cholecystectomy was introduced many decades ago by Madding, and it proved to be a safe choice for critically ill patients that needed a bailout decision [57]. In a systematic review and meta-analysis by Elshaer on difficult LC, the adoption of laparoscopic SC was associated withsuperior outcomes compared with open SC [58]. In an attempt to comprehensively research the techniques of SC, Lunevicius identified 3 reviews, 2 cohort studies, and 67 case series from 1995 onwards. None of the authors had previously tried to categorise the techniques of SC. The author of [59] identified four variants ofLSC based on the size of the GB remnant and the type of excision: circular excision, longitudinal removal of the visceral wall, fundectomy, and wedge excision. Regardless of the subvariant, we believe that in gangrenous cholecystitis, the local conditions will dictate the approach rather than the surgeon’s preference; hence, a correlation between any classification and suggested techniques would be superfluous. Closure of the cystic duct should be attempted in all cases, but there is no difference in complications between cases with a sutured GB stump compared with those left open [58,60].

## 5. Complementary Visualisation Techniques

### 5.1. Fluorescent Cholangiography

Fourteen years ago, when Ishizawa reported the first fluorescent cholangiography using intravenous indocyanine green (ICG) during an LC, he used a figure of speech that has remained relevant over the years: “a biliary road map to safe surgery” [61]. Most concretely, the delineation of invisible anatomical structures using safe substance and visual augmentation software is both a mapped dissection and a safer approach to traditional surgery. The technique spread rapidly, and numerous papers confirmed its usefulness in identifying the CBD and vascular structures covered with fatty tissues [62]. In a survey of general surgeons who analysed video vignettes with cystic pedicle dissections with and without augmented reality, fluorescence improved the recognition of structures compared with conventional imaging [63]. To date, there hasonly been one randomised control trial (RTC) produced by a multicentric team on 670 LC that has compared NIR-fluorescence-assisted LC with conventional white-light LC. Anatomical landmarks were identified three times quicker in the arm with augmented reality [64]. Since the majority of iatrogenic lesions were attributed to the misinterpretation of local anatomy, we believe that using ICG makes LC in acute cholecystitis safer (Figure 9).

### 5.2. Intravital Dye Injection into the Gallbladder 

The use of intravital dyes to obtain intraoperative cholangiography encompasses simple colorimetric methods that do not involve fluorescence or radiation. The main substances used are methylene blue and ICG. The latter has a bright green colour in conventional light and can be used as a fluorophore if the technology is available. As opposed to the NIR fluorescent cholangiography obtained through the excretion of ICG into the bile, the intravital method is instead ”cholecysto-cholangiography”, suggesting that the delivery of a dye involves a transparietal injection into the gallbladder [65,66]. Compared with venous administration of ICG, direct injection eliminates the background fluorescence of the liver and offers enhancement of Hartmann’s pouch and the cystic duct, provided that it is not occluded by severe scarring or a calculus [4,67]. Many authors have reported reduced operative times and improved visualisation using intravital dyes [68,69]. Skrabec usefully comments that in a distended GB in which evacuation is required as part of the surgical act, transparietal injection of a dye does not add to the operating time.

### 5.3. Conventional Transcysticand Direct Injection Cholangiography

Before the introduction of NIR fluoroscopy, radiologic cholangiography was the only alternative to colorimetric methods. This technique requires special training, involves a learning curve, and produces radiation for both the patient and the surgeon [70]. A systematic review and meta-analysis by Kovacz demonstrated that in a large number of cases, IOC did not ameliorate the prevalence of VBI but increased the operative times; hence, the paper confirmed the protective value against iatrogenic lesions of routine IOC with radiopaque dyes [71]. We believe that in difficult cases, identifying the cystic duct would be problematic, rendering transcystic cholangiography difficult and unreliable for VBI avoidance. The direct injection of the contrast would seem more appropriate, but the data in the literature are scarce and often reduced to case series [72,73].

### 5.4. Laparoscopic Ultrasonography 

The miniaturisation of ultrasound probes and the potential of ultrasonography for visualising anatomical structures surrounded by fatty tissues has not caused a pervasive use of the technique because of the high costs and special training requirements [74,75]. Few publications have focused on VBI prevention using laparoscopic ultrasonography (Lap USS), with a notable contribution from Gwin, who published a prospective study on patients with acute cholecystitis in which Lap USS was considered a crucial instrument in the delineation of the cystic duct and the CBD [76]. Compiled data from Dili, who discussed the implementation of Lap USS in a large systematic review, show that surgeons are reluctant to embrace this technique despite its clear advantages [77].

## 6. Cognitive Maps and Algorithms for Safer Laparoscopic Cholecystectomy

### 6.1. Sutherland’s B-SAFE Acronym

Visual representations of mental models are useful in enhancing learning and improving memory. Cognitive maps are an instrument that enablesurgeons to identify potential challenges and anticipate critical intraoperative steps. Procedural planning, spatial orientation, intra-operative time-out rules, and other debiasing methods can be coded in maps, acronyms, and schemas that are useful to even the most skilful and experienced surgeons. Sutherland wrote about the importance of cognitive instruments in preventing bile duct injuries and identified a ”slow analytic mode” and a “fast thinking mode” during gallbladder surgery (Figure 10). The author proposed an acronym for situational awareness called B-SAFE, suggesting that surgeons should employ ongoing time-out and confirmation of anatomical “safe” landmarks [78]. Schendel’s rules for defining ideal anatomical landmarks (highly prevalent, easily recognisable, and conveying accurate anatomical relations) should apply to all structures incorporated in mental maps to expand the benefit of using them [79]. Compound maps composed of multiple items are more accurate than simple ones based on an accretion effect [80].

### 6.2. The Critical View of Safety

The most cited and widely spread conceptual map for safe LC is Strasberg’s critical view of safety introduced almost three decades ago [8]. This principle is as elegant as it is simple, and is considered to be the gold standard in approaching LC from Calot’s triangle [81]. The following three steps must be achieved during LC to ensure a safe dissection: (I) the hepatocystic triangle should be cleared of fatty tissue, (II) the cystic plate should be exposed in its lower third, and (III) only two structures should enter the GB. Applying the CVS principles is even more important considering that the Calot-first approach is the standard technique, and LC is the most prevalent laparoscopic surgery in the world. However, various authors have reported that the implementation of the CVS is “lacunar”, and experienced surgeons tend to compromise on standardised dissection rather than novices [82]. There is no strong evidence to support that the use of the CVS is superior to other cognitive instruments to perform safe LC, but we believe that the concept’s popularity stems from the widespread nature of the Calot-first approach [83]. 

### 6.3. Time-Out and Second Opinion 

The SAGES six steps for a safe cholecystectomy define time-out as a “momentary intraoperative pause” to allow for reflection before “clipping, cutting or transecting any ductal structures”[84]. This invitation to increased awareness was quantified by Mascagni et al. in a study that compared the quality of achievement of the CVS before and after the introduction of a requirement to assess the quality of the CVS, called the 5 srule. The study demonstrated that after implementing the contemplative step, the quality of the CVS improved, and the number of bailout procedures increased, with more surgeons opting for safer decisions [85]. Deng considers that landmarking and time-out are closely related and that there are two main reflective moments during any LC: at the beginning of the surgery when one identifies the anatomical landmarks and during the final assessment of the CVS [86]. The cognitive and executive aspects of the surgical interventions are interlinked and can only be separated for didactic purposes; therefore, it is only natural to accept that time-out and other psychological instruments enhance overall performance. 

### 6.4. Strategic Thinking, Mind Maps, and Algorithms

Pre-operative comprehensive risk evaluation and selecting the method that best fits each patient are important elements to increase patients’ safety [87]. Decision making in any field is not just the end-point of an analytical assessment but also involves intuition, emotions, awareness, and tacit knowledge [88]. These processes act as modulators during surgery and contribute to the outcome [89,90]. Mind maps and algorithms use heuristic pathways to facilitate correct decision making and can contribute to safer practice. 

Mintzberg proposed seven frameworks of “seeing” (ahead, behind, above, below, beside, beyond, and through) for achieving the best results using strategic thinking [91]. This multifaceted approach is translated into practice using algorithms and mind maps. For instance, an algorithm for a difficult LC looks ”ahead“ by anticipating potential error traps derived from a misinterpretation of the local anatomy or deficient ergonomy, such as over-traction, and invites caution and optimal exposure. It looks “behind” by evoking past mistakes and “above” by implementing safety rules. 

The acknowledgement of VBIs triggers an increase in attention and commands an evaluation of the difficulty. Gaining exposure and orientation are preparatory steps for successfully identifying anatomical landmarks that are mandatory cues for a safe dissection (rules). Looking “below” when safety criteria are not met prompts reversion to the basic principles of “identification before cutting” and “correct exposure”. When goals are difficult to achieve, a useful algorithm looks “beside” for complementary aids such as “time out”, “reassessment”, or “second opinion”, and it looks beyond by employing additional supportive measures and alternatives to classical and conventional methods such as Lap USS, IOC, and NIR fluorescence. Finally, it looks “through” when unavoidable challenges must be tackled, suggesting bailout decisions such as different techniques, subtotal LC, or conversion (Figure 11).

When memory erodes, acronyms or mnemonic diagrams help a surgeon to remember theprinciples that must be rigorously followed during safe surgery. Oversimplification, such as the A, B, C, and D approach, ensures adherence to the most valuable concepts of safety (Figure 12).

Although knowledge and skills are the mainstays of a surgeon’s practice, acknowledging human factors and ergonomic principles in the decision-making process will, nevertheless, make it safer and more beneficial to patients.

## 7. Conclusions

Laparoscopic cholecystectomy remains the most common surgery worldwide, lending great significance to its most prevalent iatrogenic lesion: VBIs. The pervasive tendency of using minimally invasive techniques did not lead to an increase in VBIs because a culture of safe LC has independently spread from the adoption of laparoscopy. Unfortunately, there is no systematisation of teaching nor equal access to technologies and resources. This makes the management of difficult LC significantly heterogeneous and has caused many authors to address the issue and propose a myriad of solutions tailored to their society and local circumstances. There are no robust data to disqualify or enforce their recommendations for safer cholecystectomy, nor an author to exhaustively curate them. Since most injuries are caused by an erroneous interpretation of the local anatomy, easier methods for recognising intraoperative clues must be produced. The main anatomical landmarks were systematised according to their clinical significance, and based on simple anatomical principles, strategic operative concepts and various techniques were explained with hints to their most relevant indications. Additionally, visualisation aids such as Lap USS, NIR fluorescence, and IOC with radiopaque or intravital dyes have beendiscussed, culminating inhighlighting the importance of cognitive maps, algorithms, and mnemonics. These four pillars of safer LC must be consolidated with future studies that can produce irrefutable statistical data about using augmented reality and improved training methods.

## Figures and Tables

**Figure 1 medicina-59-01491-f001:**
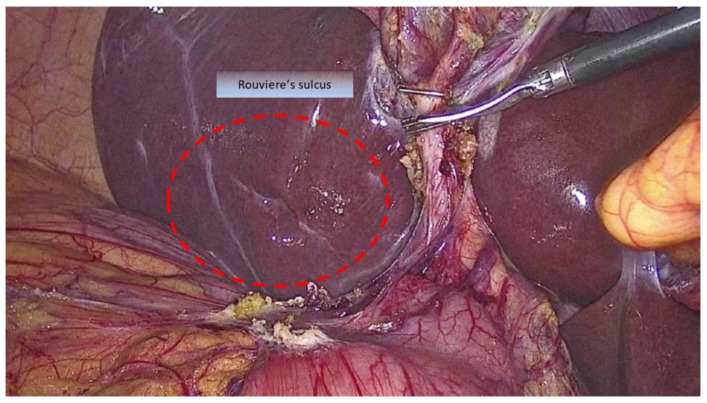
Intraoperative depiction of Rouviere’s sulcus (circumscribed by the red dotted line).A theoretical construct derived from RS is the R4U line concept introduced by Gupta [40]. The area above the plane that crosses through RS under the fourth segment of the liver and towards the umbilical ligament should be safe for dissection.

**Figure 2 medicina-59-01491-f002:**
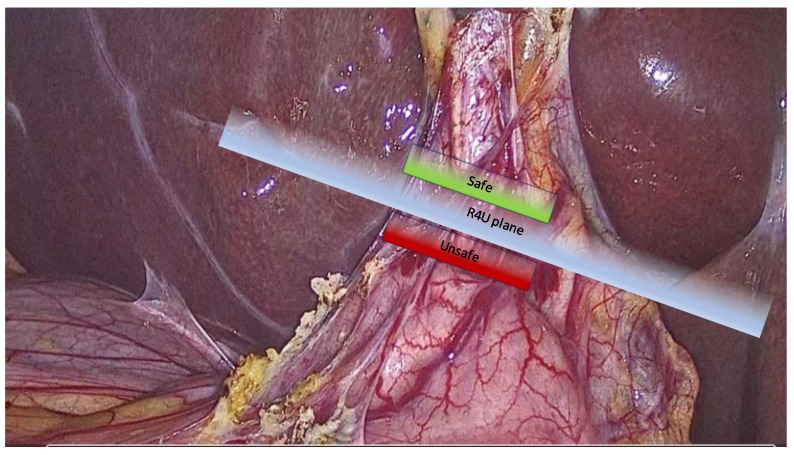
Depiction of the R4U plane (blue) and safety zones (safe plane in green, unsafe plane in red).

**Figure 3 medicina-59-01491-f003:**
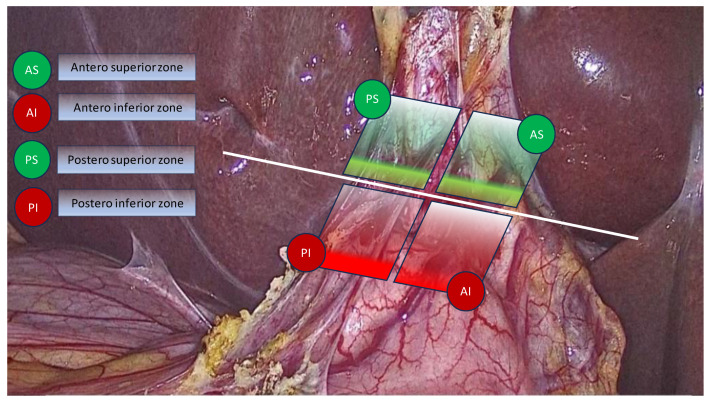
Zonal demarcation for safe dissection according to Gupta with four zones: AS—antero-superior zone, AI—antero-inferior zone, PS—postero-superior zone, PI—postero-inferior zone. Safe zones coloured in green, unsafe zones coloured in red.

**Figure 4 medicina-59-01491-f004:**
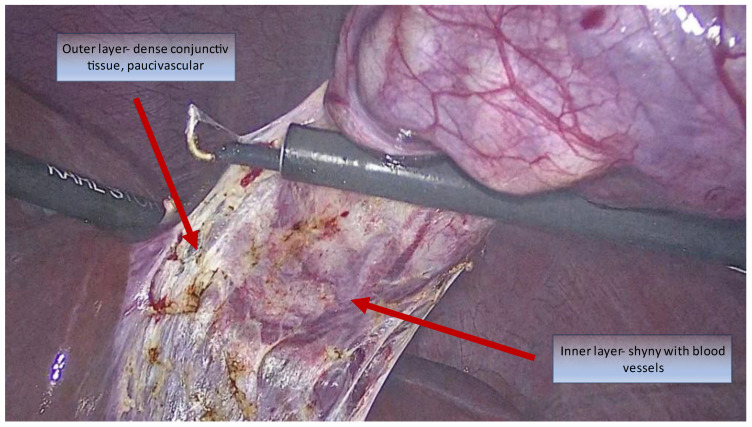
Intraoperative view of the two subserosal layers. Inner and outer layers pointed out by arrows.

**Figure 5 medicina-59-01491-f005:**
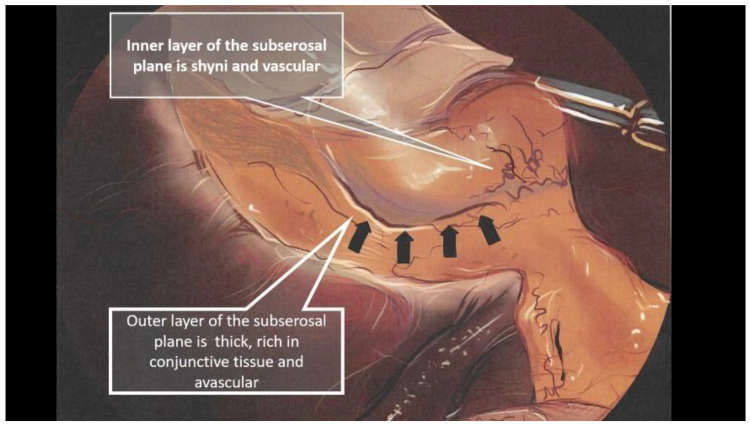
Schematic representation of the two subserosal layers modified after Honda. The arrows define the semi-circular area for the peritoneal incision and dissection of the outer SS layer that allows the visualization of the inner SS layer.

**Figure 6 medicina-59-01491-f006:**
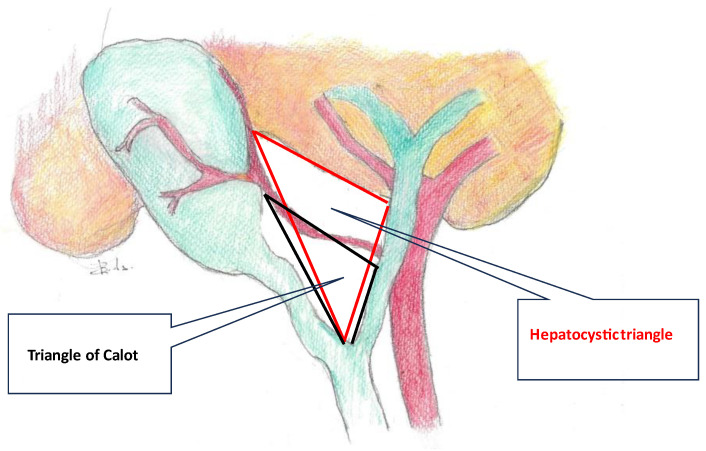
Schematic representation of Calot’s (black line) and Bode Rocko (red line) triangles.

**Figure 7 medicina-59-01491-f007:**
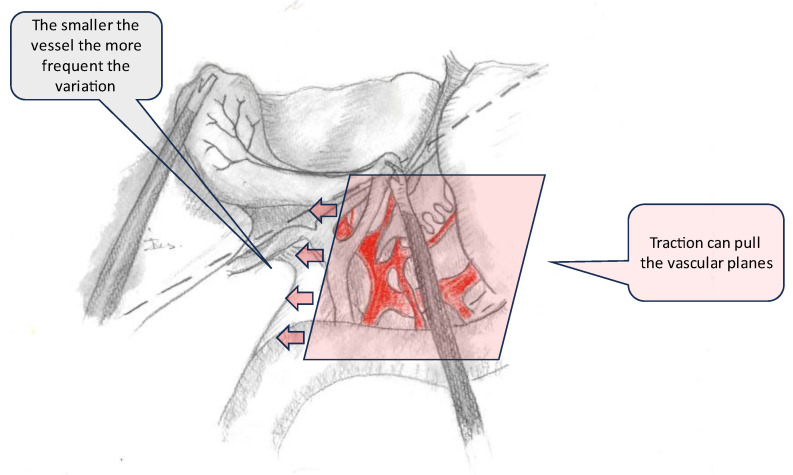
Depiction of traction direction (arrows) during exposure of Calot’s triangle with emphasis on one of the vascular structures(coloured in red).

**Figure 8 medicina-59-01491-f008:**
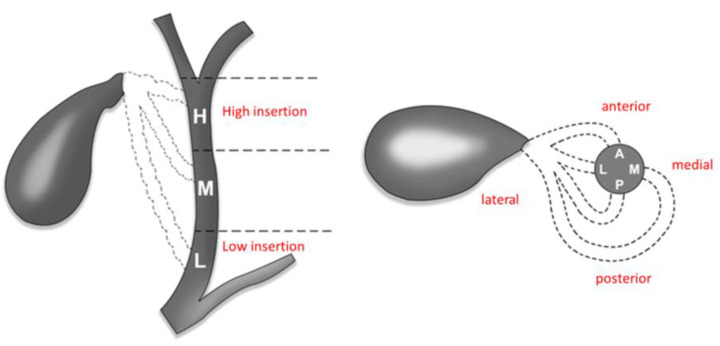
Depictions of the variations in the insertion of the cystic duct—modified after Gunduz: H—high insertion, M—medium insertion, L—low insertion. A—anterior, L—lateral, P—posterior, M—medial.

**Figure 9 medicina-59-01491-f009:**
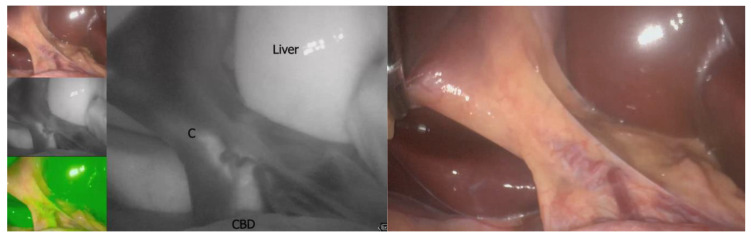
Intraoperative view of the common bile duct in conventional (**left**) and NIR light (**right**)—personal collection. C—cystic duct, CBD—common bile duct.

**Figure 10 medicina-59-01491-f010:**
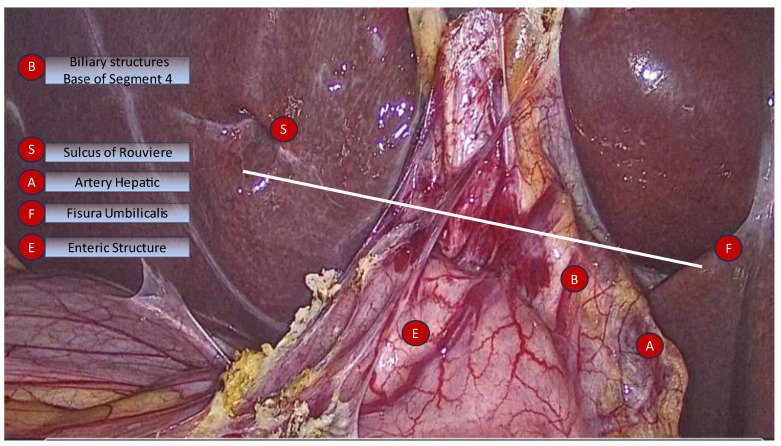
Sutherland’s B-SAFE acronym. B—biliary structures and base of segment 4, S—Rouviere’ssulcus, A—hepatic artery, F—fisuraumbilicalis, E—enteric structure. White line: R4U line.

**Figure 11 medicina-59-01491-f011:**
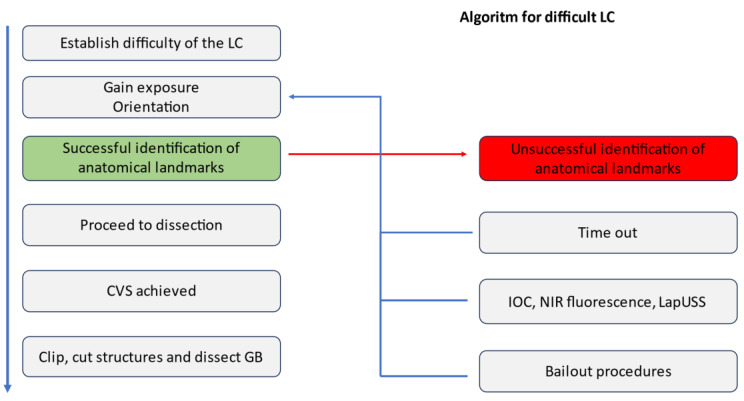
Algorithm for difficult LC based on Mintzberg’s perspectives on strategic thinking. The algorithm flow for successful identification (green) vs unsuccesful identification (red) of the anatomical landmarks.

**Figure 12 medicina-59-01491-f012:**
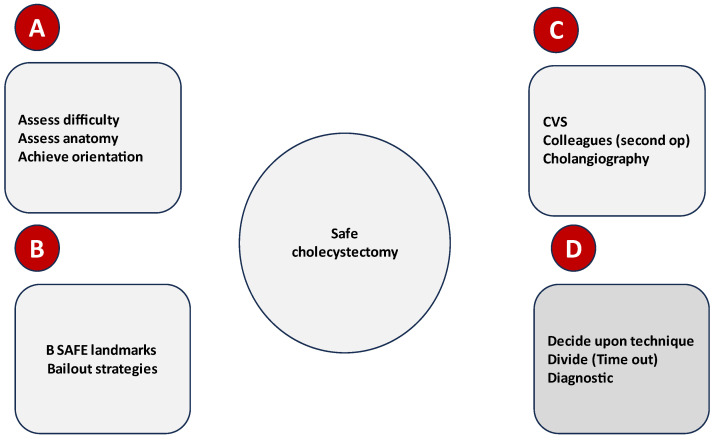
Mnemonic map:principles of safety in LC; (**A**–**D**)—represent successive steps of the mnemonic map.

**Table 1 medicina-59-01491-t001:** G10 score for assessing the difficulty of an LC [32].

Sugrue Intraoperative Severity Score	Score
**Appearance**	
● Adhesions < 50% of GB	1
● Adhesions > 50% of GB	2
● Completely buried GB	3
**Distension/contraction**	
● Distended GB or contracted shrilled GB	1
● Inability to grasp without decompression	1
● Stone of more than 1cm impacted in Hartmann’s pouch	1
**Access**	
● BMI > 30	1
● Adhesions from previous surgery	1
**Sepsis and complications**	
● Free bile or pus outside the GB	1
● Fistula	1
Total possible	10

**Table 2 medicina-59-01491-t002:** Clinical significance of main anatomical landmarks during LC.

Landmark	Clinical Significance	Expected Outcome
Rouviere’s sulcus	Delineates safety areas of dissection	Reduced VBI
R4U plane	A derivative of Rouviere’s sulcus/safe areas	Reduced VBI
Inner and outer layers	Provides a plane of dissection	Increased number of TCs Facilitation of dissection
Triangles of Calot and Budde	Provides recognisable anatomical landmarks	Standardised technique Reduced VBI
Anatomical variations	Atypical sites for VBI	Increase in awareness of atypical anatomy

Footnote: VBI = vasculo-biliary injuries.

## Data Availability

No new data were created.

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
