# Peer review of "When Critical View of Safety Fails: A Practical Perspective on Difficult Laparoscopic Cholecystectomy"

_medicina, 2023, doi:10.3390/medicina59081491_

Round 1

Reviewer 1 Report

Dear Authors

I would like to thank you for the opportunity of reviewing this interesting paper that is focused on a very remarkable and challenging topic that is a lively argument also in daily clinical practice. 

This paper is a narrative review that discusses from a contextual point of view the unmet need for standardization of the surgical approach in difficult laparoscopic cholecystectomies, the main strategic operative concepts and techniques, complementary visualization aids for delineation of anatomical landmarks and the importance of cognitive maps and algorithms in performing safer procedures.

This paper is pleasurable to read, although it suffers from some limitations that Authors can easily adjust in order to slightly improve their review making it more eligible for this important Journal. Furthermore, the Authors can improve some sections of the paper, adding information and including other important references about this topic that, in my opinion, should be cited and discussed. 

First of all, the Authors did not correctly report keywords from MeSH Browser. This is important, in my personal opinion, in order to increase the traceability of this paper (and consequently the possibility of the Journal being cited by Readers and Stakeholders). I suggest the check of all KW.

The introduction is very long and some parts must be significantly reduced or even remove, such as lines 54-57, 68-77, 79-86 and 99-111.

Furthermore, some sentences sound a bit odd (for example, “leading great significance” could be corrected with “but significantly contributes” in line 59). Therefore, although the language used is quite appropriate, I (I am not a native English speaker) recommend to Authors to obtain a certified native speaker with proficiencies in the scientific-medical field to complete properly this paper (if not yet done).

In my opinion, it is important to underline that iatrogenic bile duct injury (IBDI) secondary to cholecystectomy may crucially affect long-term quality of life and have major morbidities. Furthermore, even after reconstructive surgical treatment, such injuries still reduce the long-term quality of life [doi: 10.47717/turkjsurg.2023.5780][doi: 10.1016/j.hpb.2021.02.014]. Please briefly add these considerations in the text and cite the aforementioned references.

In line 92, “Despite being an old paper”, 2001 is not quite old, please correct.

In chapter 2 “Standardisation of techniques for emergency LC”, please mention the fundamental role of US, CT and MRI in the diagnosis, management and pre-operative study of acute cholecystitis. In fact, although the diagnostic criteria for the diagnosis of acute cholecystitis by US and its diagnostic yield vary in different studies, its low invasiveness, widespread availability, ease of use, and cost-effectiveness make it recommended as the first-choice imaging method for the morphological diagnosis of acute cholecystitis. If the abdominal US does not provide a definitive diagnosis, CT and MRI are other imaging studies that are less commonly used to diagnose acute cholecystitis. [doi: 10.1002/jhbp.515] [doi: 10.1007/s10140-021-01944-z][ doi: 10.1001/jama.2022.2350] MRI/MRCP allows an accurate study of the anatomy of the biliary system and the evaluation of any anatomical variants and/or accessory ducts, making it pivotal for preoperative investigation. The use of MRI with MR cholangiopancreatography in the emergency setting provides rapid, noninvasive, and confident diagnosis or exclusion of acute cholecystitis and of coexistent choledocholithiasis. Some previous literature reports, suggest that, when available, MRI should be recommended to provide prompt and efficient triage of patients with suspected cholecystitis and inconclusive clinical, laboratory, and sonographic findings. [10.1007/s10140-012-1038-z] [ doi: 10.1111/joa.13808]

Both MRI and CT can be also used for diagnosing gangrenous cholecystitis whereas only CT is recommended for diagnosing emphysematous cholecystitis. Please discuss these imaging techniques and their role and cite the aforementioned references.

Please remove lines 120-126, they are not necessary.

Finally, I think references in the text should be reformatted as suggested by Author’s guidelines (for example, in line 54 “1,3” should be corrected with “1-3”).

Author Response

Dear Reviewer,

Many thanks for your time spend in reviewing our manuscript and your helpful comments. We have carefully revised and updated our manuscripts according to your recommendations.

 We have revised the keywords, according to MeSH terms, as recommended.

The introduction is very long and some parts must be significantly reduced or even remove, such as lines 54-57, 68-77, 79-86 and 99-111.

Response: We have shortened the Introduction as recommended and updated the references accordingly.

Furthermore, some sentences sound a bit odd (for example, “leading great significance” could be corrected with “but significantly contributes” in line 59). Therefore, although the language used is quite appropriate, I (I am not a native English speaker) recommend to Authors to obtain a certified native speaker with proficiencies in the scientific-medical field to complete properly this paper (if not yet done).

Response: We have largely revised the manuscript for English language using MDPI English Editing service.

In my opinion, it is important to underline that iatrogenic bile duct injury (IBDI) secondary to cholecystectomy may crucially affect long-term quality of life and have major morbidities. Furthermore, even after reconstructive surgical treatment, such injuries still reduce the long-term quality of life [doi: 10.47717/turkjsurg.2023.5780][doi: 10.1016/j.hpb.2021.02.014]. Please briefly add these considerations in the text and cite the aforementioned references.

Response: WE definitely agree and thank you for the useful comment. We have added this info in the text and the suggested references.

In line 92, “Despite being an old paper”, 2001 is not quite old, please correct.

Response: We have corrected.

In chapter 2 “Standardisation of techniques for emergency LC”, please mention the fundamental role of US, CT and MRI in the diagnosis, management and pre-operative study of acute cholecystitis. In fact, although the diagnostic criteria for the diagnosis of acute cholecystitis by US and its diagnostic yield vary in different studies, its low invasiveness, widespread availability, ease of use, and cost-effectiveness make it recommended as the first-choice imaging method for the morphological diagnosis of acute cholecystitis. If the abdominal US does not provide a definitive diagnosis, CT and MRI are other imaging studies that are less commonly used to diagnose acute cholecystitis. [doi: 10.1002/jhbp.515] [doi: 10.1007/s10140-021-01944-z][ doi: 10.1001/jama.2022.2350] MRI/MRCP allows an accurate study of the anatomy of the biliary system and the evaluation of any anatomical variants and/or accessory ducts, making it pivotal for preoperative investigation. The use of MRI with MR cholangiopancreatography in the emergency setting provides rapid, noninvasive, and confident diagnosis or exclusion of acute cholecystitis and of coexistent choledocholithiasis. Some previous literature reports, suggest that, when available, MRI should be recommended to provide prompt and efficient triage of patients with suspected cholecystitis and inconclusive clinical, laboratory, and sonographic findings. [10.1007/s10140-012-1038-z] [ doi: 10.1111/joa.13808]

Both MRI and CT can be also used for diagnosing gangrenous cholecystitis whereas only CT is recommended for diagnosing emphysematous cholecystitis. Please discuss these imaging techniques and their role and cite the aforementioned references.

Response: Thank you very much for the suggestion. We have added a paragraph with the role of US, CT and IRM  in the diagnosis of the acute cholecystitis and added the recommended references.

Please remove lines 120-126, they are not necessary.

Response: We have removed this paragraph.

Finally, I think references in the text should be reformatted as suggested by Author’s guidelines (for example, in line 54 “1,3” should be corrected with “1-3”).

Response: Thank you, we have corrected it.

We hope in this revised version you will find it suitable to be published.

Reviewer 2 Report

Thank you for offering to review. Fine review of a prevalent topic. Very complete and entertaining paper, had a great time reading it, and I believe it would be of interest of many general surgeons. Congratulations.

Author Response

Dear Reviewer, many thanks for your kind words and appreciation of our work!

Reviewer 3 Report

This article is about safe laparoscopic cholecystectomy, well explained, with contributions of surgical anatomy pictures.

The article should be written in native English.

Author Response

Dear Reviewer, many thanks for your kind words and appreciation of our work! We have revised the English language extensively, using the MDPI Editing services. We have attached the proof. We hope in this revised version, you will find it suitable to be published.

Reviewer 4 Report

The review was written well  all aspect of complications  and injuries were  added 

figures  and pictures  must have  more details 

a few  editing in punctuations   and prepositions

surgical techniques for acute cases  change to  " in acute cases"

approach towards LC for acute cases  change to  " for LC  in acute cases"

ulminating with highlighting the importance of cognitive maps, algorithms and 461mnemonics We believe   change to "mnemonic . We believe"

Author Response

Dear Reviewer, many thanks for the time spending in reviewing our paper. We have done a comprehensive English editing based on your comments and using English Editing services provided by MDPI. We have added more details in Figures, as recommended. We do hope that in this revised version you will find it suitable to be published.

Round 2

Reviewer 1 Report

the authors have addressed the raised points adequately.